# An ultrasensitive molybdenum-based double-heterojunction phototransistor

Shun Feng[1,2,7], Chi Liu [1,7], Qianbing Zhu[1,3], Xin Su[4], Wangwang Qian[1,3], Yun Sun[1], Chengxu Wang[1,3], Bo Li[1,3], Maolin Chen [1,3], Long Chen[1], Wei Chen[1,3], Lili Zhang [1], Chao Zhen[1], Feijiu Wang[5], Wencai Ren [1,3], Lichang Yin[1,3 ✉], Xiaomu Wang [4 ✉], Hui-Ming Cheng [1,3,6 ✉] & Dong-Ming Sun [1,3 ✉]

Two-dimensional (2D) materials are promising for next-generation photo detection because of their exceptional properties such as a strong interaction with light, electronic and optical properties that depend on the number of layers, and the ability to form hybrid structures. However, the intrinsic detection ability of 2D material-based photodetectors is low due to their atomic thickness. Photogating is widely used to improve the responsivity of devices, which usually generates large noise current, resulting in limited detectivity. Here, we report a molybdenum-based phototransistor with $MoS_2$ channel and $\alpha$-$MoO_{3-x}$ contact electrodes. The device works in a photo-induced barrier-lowering (PIBL) mechanism and its double heterojunctions between the channel and the electrodes can provide positive feedback to each other. As a result, a detectivity of $9.8 \times 10^{16}$ cm $Hz^{1/2}$ $W^{-1}$ has been achieved. The proposed double heterojunction PIBL mechanism adds to the techniques available for the fabrication of 2D material-based phototransistors with an ultrahigh photosensitivity.

[1] Shenyang National Laboratory for Materials Science, Institute of Metal Research, Chinese Academy of Sciences, Shenyang, PR China. [2] School of Physical Science and Technology, ShanghaiTech University, Shanghai, PR China. [3] School of Material Science and Engineering, University of Science and Technology of China, Hefei, PR China. [4] National Laboratory of Solid State Microstructures, School of Physics, School of Electronic Science and Engineering, Collaborative Innovation Centre of Advanced Microstructures, Nanjing University, Nanjing, PR China. [5] Henan Key Laboratory of Photovoltaic Materials, Henan University, Kaifeng, PR China. [6] Shenzhen Geim Graphene Center, Tsinghua-Berkeley Shenzhen Institute, Tsinghua University, Shenzhen, PR China. [7] These authors contributed equally: Shun Feng, Chi Liu. ✉email: lcyin@imr.ac.cn; xiaomu.wang@nju.edu.cn; cheng@imr.ac.cn; dmsun@imr.ac.cn

Photodetectors based on two-dimensional (2D) materials usually have a low responsivity and detectivity[1–3] because atomically-thin 2D layered materials have weak light absorption. The photogating mechanism has been widely used to provide a photo gain to improve device performance[4–6], which is basically achieved either by a trap-assisted photoconductive effect[7–27] or by a photovoltaic effect[28]. For example, charges were transferred from the channel to the bound water molecules on the $SiO_2$ surface in pristine $MoS_2$ phototransistors[7], and the poor charge separation ability of water molecules leads to a relatively low detectivity. In a hybrid $MoS_2$/PbS quantum-dot photodetector, photogenerated electrons were transferred to a $MoS_2$ layer, while photogenerated holes stayed in the quantum-dots[22], and a leakage path was inevitably formed and resulted in a large dark current. An all-2D $MoS_2$ phototransistor achieved a detectivity as high as $3.5 \times 10^{14}$ cm $Hz^{1/2}$ $W^{-1}$ under a high bias voltage of 10 V, in which a $MoS_2$ P–N homojunction played the roles of charge separation and a sensitive layer[23]. Overall, a phototransistor with high responsivity, low-noise current and working at low bias voltage has not been reported.

Here, we report a molybdenum-based double-heterojunction phototransistor with a $MoS_2$ channel and two α-$MoO_{3-x}$ contact electrodes. Using the energy band combination of the $MoS_2$/α-$MoO_{3-x}$ junction, the formed double heterojunctions are able to provide a positive feedback to each other with the help of light. And we proposed a working mechanism named photo-induced barrier lowering (PIBL) for this phototransistor. As a result, a high detectivity of $1.7 \times 10^{14}$ cm $Hz^{1/2}$ $W^{-1}$ was achieved. However, because the intrinsic noise of the device is too low to be measured, the detectivity of this device is seriously underestimated. Through the reasonable estimation of the intrinsic noise, the detectivity of our device is as high as $9.8 \times 10^{16}$ cm $Hz^{1/2}$ $W^{-1}$. At the same time, this device maintains a response speed with a rise time of 95 μs and a fall time of 105 μs.

## Results

### Device design and characterization

A phototransistor with a back-gate structure was fabricated by a layer transfer method (Methods, Supplementary Fig. 1), and the channel material was a few- or multi-layer $MoS_2$ flake which was stacked on the patterned multilayer α-$MoO_3$ electrodes (Fig. 1a, b and Supplementary Fig. 2). Thermal annealing at 350 °C was carried out to introduce oxygen vacancies[29] which changed the structural and electrical properties of α-$MoO_3$ (Methods). For the resulting α-$MoO_{3-x}$ $(0 < x < 1)$, the crystal lattice constant decreased slightly due to the formation of oxygen defects (Supplementary Figs. 3–5), but the conductance of α-$MoO_{3-x}$ was dramatically increased by more than 4 orders of magnitude (Supplementary Fig. 6). Figure 1c shows the cross-section of the phototransistor that has a high-quality van der Waals heterojunction of $MoS_2$ and α-$MoO_{3-x}$ covered by a 5-nm-thick $HfO_2$ passivation layer. The thin $HfO_2$ capping layer causes the n-type doping on $MoS_2$, which narrows the interfacial energy barrier between α-$MoO_{3-x}$ and $MoS_2$ (Supplementary Fig. 7). The transfer characteristics of the α-$MoO_{3-x}$/$MoS_2$/α-$MoO_{3-x}$ phototransistor show a strong optoelectronic response using a monochromatic light source with a wavelength λ of 405 nm (Fig. 1d), which is similar for 516 and 638 nm wavelengths (Supplementary Fig. 8).

The detailed optoelectronic performance of the fabricated phototransistor is evaluated to obtain its figure of merit in Fig. 2 at $V_{DS} = 1$ V. The noise density spectral (S) as a function of frequency (f) at different gate voltage ($V_{GS}$) is shown in Fig. 2a, all these low-noise spectra exhibit a typical $1/f$ power density. As shown in Fig. 2a, the S around the OFF state ($V_{GS}$ ~ from −30 to −40 V, in blue and red) drowns with background noise (in black). It is well-known that the $1/f$ (flicker) noise is mainly dominated by fluctuations of carrier density or mobility. So that the intrinsic noise of the device can be inferred from the normalized noise power density ($S/I_{Dark}^2$) as a function of frequency. Except around the OFF state, the $S/I_{Dark}^2$ of our device is almost a certain value at different frequency (f), so we can extract the real noise which drowns with background noise using $S = (S/I_{Dark}^2) \times I_{Dark}^2$ (Supplementary Fig. 8). When the device is in the OFF state ($V_{GS} = -35.2$ V), the S of the device can be as low as $9.7 \times 10^{-32}$ $A^2$/Hz. The responsivity (R) first increases with the increase of $V_{GS}$ since the photocurrent ($I_{ph}$) increases with $V_{GS}$ (Supplementary Fig. 9), and then decreases slightly and becomes stable (Fig. 2b). The external quantum efficiency (EQE) follows the changing tendency of R with $V_{GS}$ (Fig. 2c). The highest values of R and EQE of this device in the OFF state are $1.9 \times 10^5$ A/W and $5.9 \times 10^7$% respectively, reached at $P_{in} = 0.46$ mW/$cm^2$ (Fig. 2b, c). The light–dark current ratio ($I_{DS}/I_{Dark}$) reaches a maximum value of $1.4 \times 10^7$ at $P_{in} = 4.78$ mW/$cm^2$ (Fig. 2d). The detectivity ($D^*$) increases slightly from $V_{GS} = -40$ V to $V_{GS} = -35.2$ V and reaches a peak value about $9.8 \times 10^{16}$ cm $Hz^2$ $W^{-1}$ at $P_{in} = 0.46$ mW/$cm^2$ because of the increasing of R. From $V_{GS} = -35.2$ V to $V_{GS} = 0$ V, the $D^*$ decreases dramatically for the significant increase of S. In the range of $V_{GS} > 0$ V, the $D^*$ is flat because the R and S are both stable (Fig. 2e). The $D^*$ calculated using measured S shows the same tendency with the estimated S, and reaches a peak value of $1.7 \times 10^{14}$ cm $Hz^2$ $W^{-1}$ at $V_{GS} = -30$ V (Fig. 2e, Inset). The time-dependent photo response of the device was measured at bias voltage ($V_{DS}$) of 1 V and $V_{GS} = 0$ V (Supplementary Fig. 10, Fig. 2f), the rise ($T_r$) and fall ($T_f$) time of photocurrent are measured to be 95 μs and 105 μs, respectively. The devices fabricated with different scale also show a stable high performance (Supplementary Fig. 11, Table 1).

To benchmark our device, we compared 2D material-based high-performance phototransistors using $MoS_2$[8,13,15,22–24,29–32], graphene[33], $GaS$[34], $WSe_2$[35,36], b–P[37], InSe[38], $In_2Se_3$[39], and $SnS_2$[40] comprehensively (Fig. 2g). Compared with those of reported pure[29,34,36–38] and engineered 2D materials based phototransistors using various strategies, including surface plasmon enhancement[13], charge transfer assistance[8,22–24,33,40], impurity/energy band engineering[15], negative capacitance[30], gate engineering[31,32,39], and contact-engineering[35], the $D^*$ of our device is highest among all previous results, while response speed is approaching to the highest previously reported value.

### Photo-induced barrier-lowering mechanism

To reveal the origin of the ultrahigh detectivity of our devices, we first pull out the photo response of α-$MoO_{3-x}$ itself or Ti-Au/α-$MoO_{3-x}$ junctions (Supplementary Fig. 12). Then we compared the performance of four phototransistors using a shared $MoS_2$ channel but different electrodes, marked as T1, T2, T3, and T4 (Fig. 3a). The transfer characteristics measured under dark and light conditions showed obviously different optoelectronic responses when the cathode and anode of the phototransistor were replaced by α-$MoO_{3-x}$ from Ti/Au (Fig. 3b). When one α-$MoO_{3-x}$/$MoS_2$ heterojunction formed at the cathode, $I_{DS}/I_{Dark}$ increased to more than 10 times that with a Ti/Au cathode, and for T4 using double α-$MoO_{3-x}$/$MoS_2$ heterojunctions, $I_{DS}/I_{Dark}$ dramatically increased more than 1000 times. The output characteristics ($I_{DS} - V_{DS}$) also indicate that $I_{DS}$ increased for the device using one heterojunction as the cathode compared with the metal electrode case, while using double heterojunctions increased $I_{DS}$ significantly (Supplementary Figs. 13, 14). On the basis of the scanning photocurrent maps (Fig. 3c), it is found that photon absorption occurs at the α-$MoO_{3-x}$/$MoS_2$ heterojunction near the cathode.

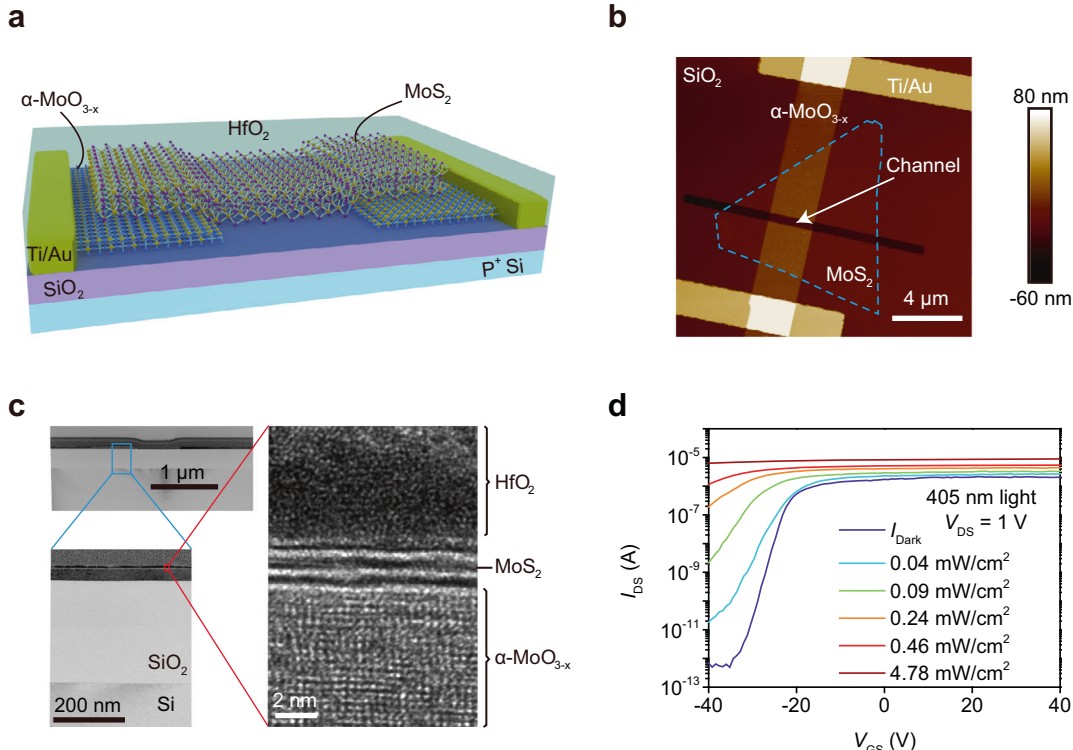

**Fig. 1 The molybdenum-based phototransistor. a** Schematic of the $\alpha$-MoO$_{3-x}$/MoS$_2$/$\alpha$-MoO$_{3-x}$ phototransistor. Using HfO$_2$ as capping layer, MoS$_2$ as channel material, $\alpha$-MoO$_{3-x}$ as contact electrodes, Ti/Au as metal contact, SiO$_2$ as dielectric layer and P$^+$ Si as gate electrode. **b** Atomic force microscope (AFM) image of the device. The dash line is ouline of MoS$_2$ channel. **c** Cross-sectional transmission electron microscope (TEM) image of the phototransistor showing the high-quality interface between $\alpha$-MoO$_{3-x}$ and MoS$_2$. **d** Transfer characteristics ($I_{DS} - V_{GS}$) of the device at $V_{DS} = 1$ V using incident light with a wavelength of 405 nm and increasing power density $P_{in}$ at room temperature. $P_{in} = P\pi^{-1}r^{-2}$, where $P$ is the actual laser output and $r$ is the radius of the laser spot (about 500 μm).

Based on the above results, a double-heterojunction PIBL mechanism was proposed to explain the ultrahigh detectivity of the $\alpha$-MoO$_{3-x}$/MoS$_2$/$\alpha$-MoO$_{3-x}$ phototransistor. Two heterojunctions exist between the channel and electrodes in the device, namely a source heterojunction ($\alpha$-MoO$_{3-x}$/MoS$_2$) and a drain heterojunction (MoS$_2$/$\alpha$-MoO$_{3-x}$). The relative positions of the conduction band minimum ($E_c$) and the valence band maximum ($E_v$) of $\alpha$-MoO$_{3-x}$ and MoS$_2$ are critical to analyze the energy band structure of the heterojunctions, which are determined by experimental characterizations such as absorbance spectrum, ultraviolet photoelectron spectroscopy (UPS) and photoluminescence spectrum (PL) (Fig. 4a, b and Supplementary Fig. 15) and theoretical techniques (Supplementary Fig. 16). Bandgaps of $\alpha$-MoO$_3$ and $\alpha$-MoO$_{3-x}$ are 3.28 and 3.24 eV, respectively, were obtained from the absorption spectra (Fig. 4a). The work function of $\alpha$-MoO$_{3-x}$ was obtained from the cutoff region, which is 5.32 eV. The difference between the Fermi level and the valence band maximum was calculated to be 3.05 eV from the valence region (Fig. 4b). Particularly, Fig. S15 shows the PL spectrum (red line) of the $\alpha$-MoO$_{3-x}$ flakes. The peak at 560 nm induced by oxygen vacancies determines the defect band is 2.21 eV above valence band maximum ($E_v$), corresponding to the density function theory (DFT) calculations (Supplementary Fig. 16), which is responsible for the 4 orders-of-magnitude increase in the conductance of $\alpha$-MoO$_{3-x}$ after annealing (Supplementary Fig. 6). Besides, compared with the MoS$_2$ flakes, the luminescent intensity of the $\alpha$-MoO$_{3-x}$/MoS$_2$ heterojunction is significantly reduced, which means that excitons are efficiently dissociated at the $\alpha$-MoO$_{3-x}$/MoS$_2$ interface (Supplementary Fig. 15, black and green lines).

If combined with the well-known MoS$_2$ band structure obtained by spectroscopic measurements[40] (Supplementary Fig. 17), the band

diagram of $\alpha$-MoO$_{3-x}$/MoS$_2$ heterojunction can be determined to clarify the mechanism of the photo-carrier induced feedback effect. In the dark, two n–n+ junctions are formed at the vdW heterointerface. The alignment of the Fermi level leads to charge transfer between $\alpha$-MoO$_{3-x}$ and MoS$_2$, which naturally results in two space-charge regions. Our device is generally operated at OFF state of the phototransistor, where MoS$_2$ is almost fully depleted. At this region, MoS$_2$ present very small density of states. Figure 4c shows a schematic of the energy band of $\alpha$-MoO$_{3-x}$/MoS$_2$/$\alpha$-MoO$_{3-x}$ when a positive bias is applied to the drain of the phototransistor in the dark. The major source of dark noise of our device is flicker noise (Fig. 2a, Supplementary Fig. 8), which is caused by carrier density fluctuations attributed to traps associated with contamination and crystal defects. The electrons at the source side faces a large Schottky barrier at a low gate voltage which blocks the traps with the energy below the barrier. We have quantitatively characterized the Schottky barrier height as about 0.55 eV at $V_{GS} = -36$ V in dark (Fig. 4d, Supplementary Fig. 18a–f, m), which is five to ten times than the Ti contact case[41,42], resulting in such low flicker noise (around $10^{-29}$ A$^2$/Hz, Supplementary Fig. 11). As shown in process (1) in Fig. 4e, by light illumination, electron-hole pairs are dynamically generated at the reversely biased source junction and subsequently separated by the built-in filed. Or in other words, there is an equivalent charge transfer between $\alpha$-MoO$_{3-x}$ and MoS$_2$, which rapidly modifies the Fermi level at the source and leverages the carrier density. The process causes an increase in the $E_F$ of the source and a decrease in the barrier at the source heterojunction, making it much easier to inject electrons from the source. In the MoS$_2$ channel, the injected electrons will raise the concentration of charge carriers, which narrows the space-charge region and lowers the barrier at the drain side. As a feedback of this barrier lowering, more voltage drops

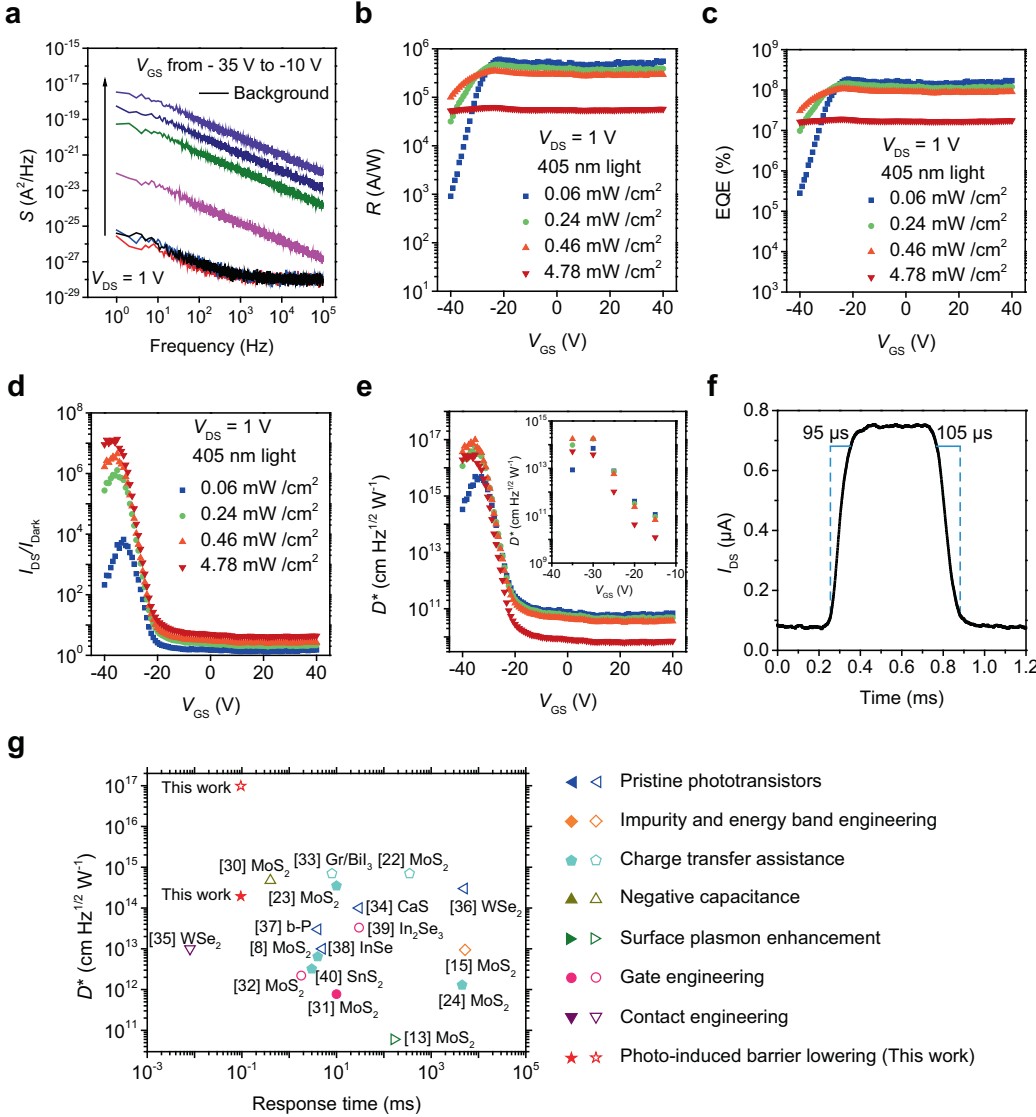

**Fig. 2 Optoelectronic performance of the phototransistor. a** Density spectral ($S$) as a function of frequency at $V_{DS} = 1\,V$ with different gate voltage $V_{GS}$ of $0\,V$, $-15\,V$, $-20\,V$, $-25\,V$, $-30\,V$, and $-35\,V$ as well as the background noise from top to down. **b** Responsivity ($R$) as a function of $V_{GS}$. $R = (I_{DS} - I_{Dark})/P_{in}$, where $P_{in}$ is power density of light. **c** $V_{GS}$-dependent external quantum efficiency (EQE). $EQE = hcR\lambda^{-1}e^{-1}$, where $h$ is the Planck constant, $c$ the speed of light, $\lambda$ the wavelength of light (405 nm) and $e$ the electron charge. **d** $V_{GS}$-dependent light–dark current ratio ($I_{DS}/I_{Dark}$). **e** $V_{GS}$-dependent detectivity ($D^{\star}$), which was calculated using intrinsic $S$. $D^{\star} = (AB)^{1/2}R/S^{1/2}$, where $A$ is the active area of ~2.5 μm², $B$ is the bandwidth (1 Hz). Inset: $V_{GS}$-dependent $D^{\star}$, which was calculated using measured $S$. **f** Response speed of the phototransistor showing the rise time of 95 μs and the fall time of 105 μs. **g** Benchmark of the phototransistor in this work demonstrating a high $D^{\star}$ compared to previously reported 2D materials based photodetectors. The $D^{\star}$ represented by hollow polygon were calculated by estimated noise, and represented by solid polygon were calculated using measured noise.

at the Source junction. Therefore, the double heterojunction enables a positive feedback to each other in one phototransistor. We also have quantitatively characterized the Schottky barrier height as about 0.042 eV at $V_{GS} = -36\,V$ under a 405 nm light ($P_{in} = 0.6\,mW/cm^2$) (Fig. 4f, Supplementary Fig. 18g–l, n), which is much lower than that in dark, resulting in such high responsivity (around $10^5$ A/W, Supplementary Fig. 11). The changing of band structure of our device from dark to light is responsible for the record-high detectivity (ranging from $10^{15}$ to $10^{17}$ cm Hz$^{1/2}$W$^{-1}$, Supplementary Fig. 11). At the same time, our device can be intrinsically fast, because different from the general photogating mechanism using traps, it employs feedback mechanism without trapping of photogenerated carriers. At last but not least, it is worth of mentioning that the response speed may be heavily underestimated. Considering the non-optimized contact of the device and large parasitic impedance of the experimental set-up, the measured speed

(~10 kHz) approaches the limitation of the facility. A faster response time can be expected in device with RF GSG contact.

We have also explained the mechanism for the MoS₂ phototransistors without and with only one heterojunction (T1–T3 in Fig. 3a) using a schematic of the energy bands (Fig. 4g–i). When a Ti/Au contact is used, an electron Schottky potential barrier is expected for the Ti/Au-MoS₂ junction. Comparing with vdW junction, metal junction presents obvious Fermi-level pinning effect[43]. Due to the strong pinning effect, the transfer of photogenerated carriers can not lower the barrier at the Source and Drain contact. The weak photo response mainly comes from the MoS₂ itself (Fig. 4g). When α-MoO₃₋ₓ is used as the anode in T2, there is also no electron injection at the Source junction. The photogenerated electrons of MoS₂ itself will raise the concentration of charge carriers slightly, leading to a hardly turned-on Drain heterojunction. Barrier at the drain α-MoO₃₋ₓ

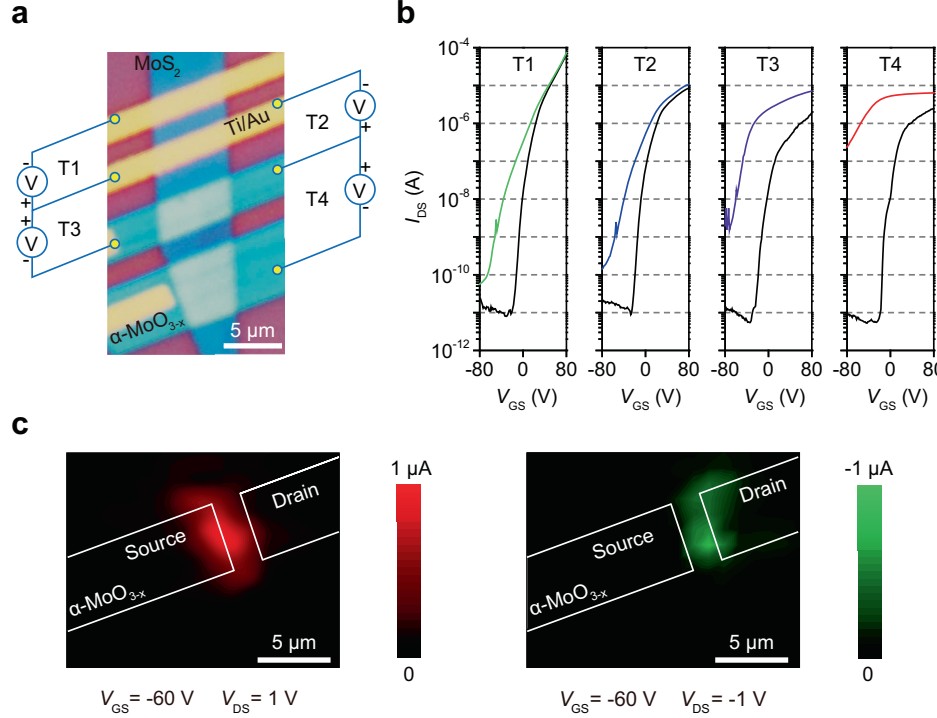

**Fig. 3 The origin of the ultrahigh detectivity. a** Four $MoS_2$ phototransistors with difference source (cathode) and drain (anode) electrodes marked T1 (Ti/Au, Ti/Au), T2 ($\alpha$-$MoO_{3-x}$, Ti/Au), T3 (Ti/Au, $\alpha$-$MoO_{3-x}$), and T4 ($\alpha$-$MoO_{3-x}$, $\alpha$-$MoO_{3-x}$). **b** $I_{DS} - V_{GS}$ measured in the dark (black line) and the light with a wavelength of 405 nm and a power density of $0.1\,mW/cm^2$ (colored line) for $V_{DS} = 1\,V$. The optoelectronic responses from T1 to T4 gradually increase, showing that $I_{DS}/I_{Dark}$ increases from <10 to more than $10^4$ at $V_{GS} = -80\,V$. **c** Scanning photocurrent map of the $\alpha$-$MoO_{3-x}$/$MoS_2$/$\alpha$-$MoO_{3-x}$ phototransistor illuminated by a 633 nm laser with a spot size of ~2 µm using a nano-positioning stage with a 1 µm scanning resolution, for $V_{GS} = -60\,V$ and $V_{DS} = \pm 1\,V$, the color bars represents quantity of photocurrent.

contact is only slightly lower than for Ti/Au in T1, resulting in a slightly better photo response (Fig. 4h). When $\alpha$-$MoO_{3-x}$ is used as the cathode in T3, barrier lowering at Source appears, resulting in a large electron injection and a large photo gain. Due to the strong pinning effect, the injected photocurrent can not lower the barrier at the Drain contact. The Source built-in field is thus invariant and no feedback occurs (Fig. 4i).

## Discussion

In conclusion, we have designed and fabricated a molybdenum-based phototransistor with one $MoS_2$ channel and two $\alpha$-$MoO_{3-x}$ contact electrodes. A double-heterojunction PIBL mechanism is proposed, in which double heterojunctions enable positive feedback to each other in one phototransistor, leading to a high detectivity of $9.8 \times 10^{16}\,cm\,Hz^{1/2}\,W^{-1}$. A fast response speed was also achieved, because our device employs the PIBL mechanism without trapping photogenerated carriers and can be intrinsically fast. Based on this mechanism, a series of 2D material-based phototransistors with high performance can be expected since the 2D material family keeps growing to enable various energy band combinations, and van der Waals heterojunctions are typically free of a lattice mismatch. 2D material phototransistors with double-heterojunction PIBL mechanism provide new technologies and shed light on the fabrication of high-performance 2D photodetectors.

## Methods

**Preparation of $\alpha$-$MoO_3$ flakes**. Bulk $\alpha$-$MoO_3$ crystals were grown by chemical vapor deposition in air environment. Commercial $MoO_3$ powder (Alfa Aesar, 99.95%, metals basis) was placed in a quartz boat that was put in the center of a horizontal tube furnace (Lindberg Blue M, TF55035KC-1). The furnace was heated to 750 °C at a rate of 25 °C/min and this temperature was maintained for 60 min until the $MoO_3$ powder was completely volatilized. When the furnace cooled to room temperature, the bulk $\alpha$-$MoO_3$ was synthesized on both ends of the quartz

tube. The bulk $\alpha$-$MoO_3$ crystals were exfoliated using Scotch® tape and multilayer $\alpha$-$MoO_3$ flakes were placed on the surface of a 290-nm-thick $SiO_2$ layer grown on a heavily p-doped silicon ($p^+$) wafer.

**Patterning of the $\alpha$-$MoO_3$ electrodes**. A polymethyl methacrylate (PMMA) layer (495k MW, A4, MicroChem) was spin-coated at 2000 rpm/min on the substrate and baked at 190 °C for 5 min, another PMMA layer (950 MW, A2, MicroChem) was then spin-coated at 4000 rpm/min and baked at 190 °C for 2 min. An undercut was created by electron-beam lithography (EBL) and developing processes. Subsequently, source and drain electrodes of the $\alpha$-$MoO_3$ flakes were patterned using reactive ion etching (RIE) ($CHF_3$ with a flux rate of 20 sccm; $O_2$ with a flux rate of 4 sccm; pressure, 2.0 Pa; power, 100 W; etching time, 1 min, see Supplementary Fig. 20) and lift-off.

**Transfer of the $MoS_2$ flake**. Polydimethylsiloxane (PDMS) was used as the medium to transfer the $MoS_2$ flakes onto the target $\alpha$-$MoO_3$ electrodes. The PDMS was prepared by stirring a mixed solution of the base and its curing agent (10: 1 in weight) and baking at 65 °C for 6 h. Few-layer $MoS_2$ flakes were exfoliated from the bulk $MoS_2$ crystals using Scotch® tape, and transferred onto the PDMS substrate, and finally released onto the target $\alpha$-$MoO_3$ electrodes using a home-made electronic van der Waals transfer station.

**Vacuum annealing of the fabricated stacks**. To produce the transition from $\alpha$-$MoO_3$ to $\alpha$-$MoO_{3-x}$, the fabricated stacks were annealed in a vacuum. The number of oxygen defects in the crystal could be controlled by the temperature and time. The annealing furnace was heated to 350 °C from room temperature in 30 min, where the temperature was maintained for 120 min. The samples were removed after the furnace cooled to room temperature. This process has no significant effect on $MoS_2$ (Supplementary Fig. 21).

**Device fabrication**. Multilayer $\alpha$-$MoO_3$ flakes were exfoliated onto a $SiO_2$/Si substrate and patterned by RIE. A few-layer $MoS_2$ flake was exfoliated onto a PDMS substrate, and then transferred onto a $\alpha$-$MoO_3$ electrode. The stack was then vacuum annealed at 350 °C for 120 min. Afterwards, metal contacts (Ti/Au: 5/50 nm) were formed by EBL, electron-beam evaporation and lift-off processes. The device was finally passivated by a 5-nm-thick $HfO_2$ layer deposited by atomic layer deposition (ALD).

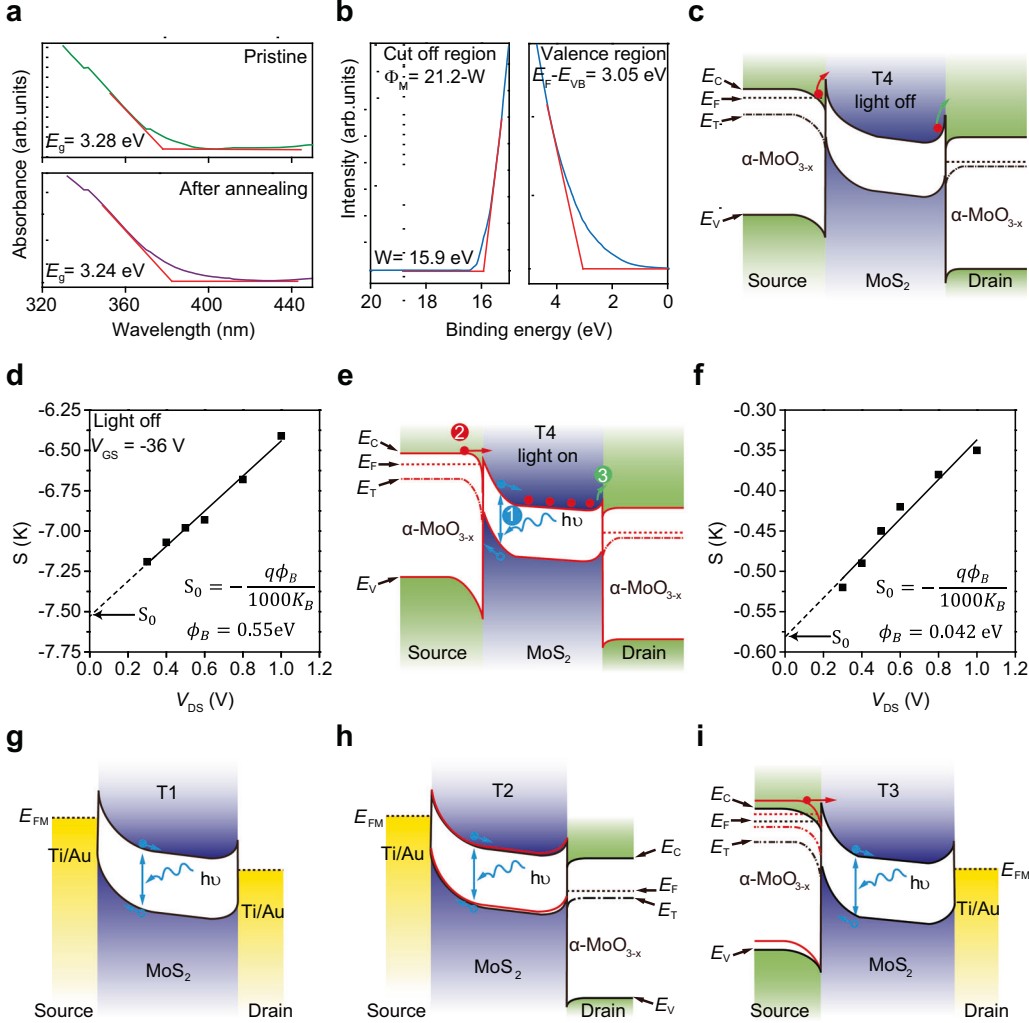

**Fig. 4 Double-heterojunction photo-induced barrier-lowering mechanism. a** Absorbance spectrum of the pure and annealed α-MoO$_{3-x}$ flakes. $E_g$ is the bandgap. The red line is an auxiliary line for extracting $E_g$. **b** Ultraviolet photoelectron spectroscopy (UPS) measurement of α-MoO$_{3-x}$ flakes. A He-I UV light source ($h\nu = 21.2$ eV) was used to obtain the UPS spectra. The binding energy is referred to the Fermi level. The work function ($\Phi_M$) was obtained from the cutoff region, which is 5.32 eV. The difference between the Fermi level and the valence band maximum was calculated to be 3.05 eV from the valence region. The red lines are auxiliary lines. **c** Energy band diagram at the light-off state. $E_C$ is the conduction band minimum, $E_F$ is the Femi level, $E_T$ is the defect band and $E_V$ is the valence band maximum. The red circle represents electrons. The red arrow represents the electron inject from source electrode to channel and green arrow represents the electron inject from channel to drain electrode. **d** Slopes extracted from Supplementary Fig. 18m as a function of $V_{DS}$. $\phi_B$ is the Schottky barrier about 0.55 eV, derived from the y-intercept, $S_O$. **e** Energy band diagram at the light-on state to illustrate the photo-induced barrier-lowering mechanism. The blue symbol (1) represents the photo-generation of electron-hole pairs, the red symbol (2) the holes and electrons separating process to reduce barrier and increase electrons injection, and the green symbol (3) the injection electrons reducing the barrier at drain side. The $h\nu$ represents incident photon. **f** Slopes extracted from Supplementary Fig. 18n as a function of $V_{DS}$. $\phi_B$ is about 0.042 eV, derived from the y-intercept, $S_O$. **g** Energy band diagram of device T1. When there is incident light, the photo response mainly comes from the MoS$_2$ material itself. $E_{FM}$ is the Femi level of Ti. **h** Energy band diagram of device T2. When there is incident light, there is a slightly better photo response mainly due to barrier-lowering at the Drain α-MoO$_{3-x}$. $E_{FM}$ is the Femi level of Ti. **i** Energy band diagram of device T3. When there is incident light, a large photo gain was achieved because Source barrier-lowering enhanced electron injection. $E_{FM}$ is the Femi level of Ti.

**Characterization**. The materials and devices were characterized using an optical microscope (Nikon ECLIPSE LV100ND), an AFM (Bruker Dimension Icon), a TEM (Thermo Scientific, Titan Cube Themis G2, acceleration voltage of 80 kV), an X-ray photoelectron spectroscopy (XPS) analyzer (Thermo VG Scientific ESCA-LAB250), a micro-Raman analyzer (Jobin Yvon HR800 using 532 nm laser excitation with a laser spot size of about 2 μm), an ultraviolet photoelectron spectroscopy (UPS) analyzer (Thermo ESCALAB 250Xi with a monochromatic Al Kα X-ray source) and a UV-Vis-NIR spectroscope (Varian Cary 5000). The electrical and optoelectronic performances were measured using a semiconductor analyzer (Agilent B1500A), a probe station (Cascade M150) and a laser diode controller (Thorlabs ITC4001, with laser excitations of 405, 516, and 638 nm) in a dark room at room temperature. The optoelectronic mapping was carried out using two semiconductor analyzers (Keithley 2400) and a micro-Raman spectroscope. To characterized the response speed of this device, a 532 nm incident laser (MDL-III-785L) modulated at various frequencies using an optical chopper was used to

illuminate the device. A current amplifier (Model SR570) was used to provide a bias voltage for the device, and an oscilloscope (Tektronix MDO3102) was used to pick up the signal. The noise was measured by a noise measurement system (PDANC300L and Fs Pro, 100 kHz bandwidth), the sketch of the noise measurement system is shown in Supplementary Fig. 22.

**Density functional theory (DFT) calculations**. DFT calculations were performed to study the electronic structure of the α-MoO$_{3-x}$/MoS$_2$ heterojunctions using the Vienna ab-initio simulation package[44]. The detailed computational settings as well as the constructions of the α-MoO$_{3-x}$/MoS$_2$ heterojunction are very similar to those used in ref. [40]. In order to obtain the accurate electronic structures, especially the bandgap values of semiconducting α-MoO$_{3-x}$ and MoS$_2$, the hybrid Heydt-Scuseria-Ernzerhof (HSE) exchange-correlation functional[45,46] was used. A consistent

screening parameter of μ = 0.2 Å⁻¹ was used for the semilocal exchange as well as for the screened nonlocal exchange as suggested for the HSE06 functional[47].

## Data availability

The data that support the findings of this study are available at https://zenodo.org/record/4835973#.YLDXJ43is2w.

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

## Acknowledgements

This work is supported by National Natural Science Foundation of China (No. 51272256, 61422406, 61574143, 51532008, 61704175, 51502304, and 51972312), the Strategic Priority Research Program of Chinese Academy of Sciences (Grant No. XDB30000000), the Key Research Program of Frontier Sciences of the Chinese Academy of Sciences (No. ZDBS-LY-JSC027), Liaoning Revitalization Talents Program (No. XLYC1807109), the Thousand Talent Program for Young Outstanding Scientists, the National Key Research and Development Program of China (2016YFB0401104), the Shandong Natural Science Foundation of China (No. ZR2019ZD49), and the projects supported by Shenyang National Laboratory for Materials Science, Institute of Metal Research, Chinese Academy of Sciences and State Key Laboratory of Luminescence and Applications, Chinese Academy of Sciences (L2019F28, Project Young Merit Scholars, SKLA-2019-03). The theoretical calculations in this work are performed on TianHe-1(A) at the National Supercomputer Center in Tianjin and Tianhe-2 at the National Supercomputer Center in Guangzhou.

## Author contributions

H.C. and D.S. conceived the idea and supervised the project. S.F. and C.L. were equal major contributors to this work. S.F. performed the device fabrication and carried out electrical and optoelectronic characterizations. S.F. and X.S. carried out the response speed and noise current density characterizations. C.L. and X.W. proposed the mechanism of the device assisted by Q.Z., F.W., M.C., W.Q., and L.Y. L.Y. carried out the theoretical calculations. Y.S. and B.L. carried out the ALD depositions. S.F., L.Z., and C.Z. were responsible for characterizing the materials. L.C. carried out the α-MoO₃ growth supervised by W.R. S.F. and C.W. carried out vacuum annealing supervised by W.C. S.F., C.L., X.W., and D.S. wrote the paper. All authors discussed the results and commented on the paper.

## Competing interests

The authors declare no competing interests.

**Additional information**

