## [Peer Review File · Nature Communications]

REVIEWER COMMENTS

Reviewer #1 (Remarks to the Author):

I would like to take this opportunity to congratulate all the authors for the significant improvement of this revised manuscript. Most of major concerns in my previous comments have been comprehensively addressed. Specifically, the proposed double-heterojunction PIBL mechanism in α -MoO_{3-x}/MoS₂/ α -MoO_{3-x} phototransistors is experimentally supported by the temperature-dependent transport measurements as well as the controlled photocurrent measurements on devices with different electrodes. As a result of this novel mechanism, a record-high detectivity of 9.8×10^{16} Jones with a reasonably ultrafast photoresponse in 2D phototransistors has been achieved. The devices were systematically characterized at different gate voltages, light wavelengths, and incident power, and the statistical analysis further confirms the stable nature of device performance. As a complement, I would like to raise some minor points needed to be further addressed, and thus recommend the publication of this work in high-impact journals such as Nature Communications.

1. The qualitative description of PIBL process in Page 11 (line 9-15) is still unclear. The authors should explain why the separation of photo-generated charge carriers at source heterojunction can lead to the increase of source Fermi-level (actually the increase of entire band), is it due to the accumulation of photo-generated holes at source interface? In addition, the decrease of energy barrier at source interface is not explicitly demonstrated in Fig. 4e and i, where the barrier height seems to remain or even increase due to the larger band bending.

2. In addition to the band structures of α -MoO_{3-x} and MoS₂, the n-type doping of MoS₂ channel by HfO₂ capping layer is also important to narrow the interfacial energy barrier for electron injection (Supplementary Fig. 7), which should be highlighted in the main text, e. g. Page 10, paragraph 2.

3. The extraction of noise power density S in the dark around the off state ($V_G \sim$ from -30 to -40 V) is not clearly demonstrated in the main text Page 5 and SI Supplementary Fig. 9. In general cases, S should be experimentally determined by the noise measurements in Supplementary Fig. 8 g. Does the authors mean that the real S of the device around off state is covered by the background noise, which can be further extracted by the proportional relation of S to I_2 for flick noise? The authors should carefully depict how to extract the S from Supplementary Fig. 8 g and h to obtain Fig. 2a, since S is the key parameter for determining the detectivity. In addition, the Fig. 8g,h and Fig. 9 in SI seems to present the same data for two different devices. It is recommended to remove Fig. 9 and add the corresponding discussion for Fig. 8g and h.

4. The language of this manuscript is required to be polished to improve the readability for a broad audience. Several expressions are not sufficiently accurate and concise for NPG publications, and a lot of spelling mistakes and grammar errors have been found throughout the manuscript. For example:

Page 2 (line 8): "Here, we report an (ultrasensitive) molybdenum-based";

Page3 (line3): "pure atomically thick 2D layer materials" should be revised to "atomically-thin 2D layered materials";

line 8 and 11: "(and) however";

Page 5 (line 14): “form”, Page 6 (line 1): “deceases”;

Page 8 (line 13 and 14): “increased” and “increases” (tense should be consistent);

Page 10 (line 2): “experimental data such as absorbance spectrum” should be revised to “experimental characterizations such as absorption spectroscopy”;

Page 11 (line 5): “which blocks the traps whose energy is below the barrier” should be revised to “which blocks the traps with the energy below the barrier”;

Fig. 4b: “3.05 (eV)”.

SI Page 11 (line 14): “indicating that the photocurrent does not determined by external” should be revised to “indicating that the photocurrent is not determined by external”.

5. Several relevant references for photogating mechanism should be added into the introduction:

1) Adv. Opt. Mater. 8, 1901971 (2020)

2) Adv. Sci. 7, 2002393 (2020)

Reviewer #2 (Remarks to the Author):

In the manuscript “An ultrasensitive molybdenum-based double-heterojunction phototransistor”, the authors present a high-performance photodetector based on MoS₂ flakes contacted through two MoO₃ electrodes, and account for the high performances on the basis of a novel mechanism involving the lowering of Schottky barriers under light irradiation. The proposed mechanism is in agreement with all the presented data, so it appears reasonable. In the present version, I particularly appreciate the Supplementary figures summarizing the performances of the different devices (Supp Fig. 12 and Supplementary Table1).

I find a major issue in the manuscript, in the extraction of the phototransistor performance. The authors do measure the spectral noise at the different back gate voltages, but they do not use the experimentally measured data to evaluate the detectivity. By contrast, they try to estimate the “intrinsic” noise level and use it for the calculation of the detectivity.

The details of how the intrinsic value is extracted are not completely clear to this reviewer, as the authors only mention “The noise density spectral (S) as a function of gate voltage (V_{GS}) at $f = 1$ Hz is shown in Fig. 2a, which is inferred from the normalized noise power density (S/I_{dark}^2) as a function of frequency” (pag 5, lines 13,14).

In my understanding, the authors notice that S is proportional to I^2 (in dark) at high gate voltages V_G , and make the assumption that the same S vs I^2 holds for low V_G , where S is below the instrumental sensitivity. Therefore, they “infer” the intrinsic S at low V_G by multiplying one of the (S/I^2) traces measured at high V_G by the dark current IDS measured at low V_G . By doing this, they obtain the graph shown in the response to the referee (Fig. R4d, and not shown in the last version of manuscript or SI), where S decreases from $10^{(-30)}$ A²/Hz at 1 Hz to $10^{(-35)}$ at high frequency.

The underlying assumption that the spectral shape of S does not depend on V_G is incorrect. Indeed, the graph shown in R4d does not make sense, since at high frequency the noise level is below the intrinsic shot noise, which I estimate to be approx 10^{-32} or 10^{-33} A²/Hz (frequency independent).

(Question: have I understood correctly how the "intrinsic" noise was estimated? The authors should describe this procedure with more details, and show the figure R4d in SI, with the caveat that the Fig. R4d would only be valid at low frequencies.)

The authors should calculate the detectivity using the measured S value, highlighting how the (measured) detectivity underestimates the real device detectivity. The inferred detectivity value (as calculated now) could be provided as an estimation of the intrinsic value, but the measured value should be also given in abstract, introduction, Fig 2a and Fig. 2e.

In any case, even considering the measured detectivity, the performances compare positively with previous works (their star would remain in the top left corner of Fig. 2g, although closer to the cloud of points). As a notice, in Fig. 2g several works estimate S as the shot noise; so the overestimation is even worse than in the present case. Perhaps the authors could plot in different ways the works in which the noise is measured or estimated, and perhaps they could add for their case both the measured and the estimated value – so the comparison could be “fair”.

And as another minor point, the authors now show the temperature dependence of the IV traces, from which they extract the Schottky barrier height. It is not clear whether this dataset was measured for a MoO/MoS₂/metal or MoO/MoS₂/MoO device; but they treat the data as if there was only one Schottky barrier, which goes against the mechanism put forward to account for the measured data. I suppose that this could still be a good approximation, considering that most of the effect comes from one of the two interfaces; the authors might want to comment on this.

Overall, in my opinion, the data presented in this work could be suitable for Nature Communications, but the performance extraction should be carefully clarified before I can recommend the acceptance of this work.

Responses to the comments of reviewers

Reviewer #1 (Remarks to the Author):

I would like to take this opportunity to congratulate all the authors for the significant improvement of this revised manuscript. Most of major concerns in my previous comments have been comprehensively addressed. Specifically, the proposed double-heterojunction PIBL mechanism in α -MoO_{3-x}/MoS₂/ α -MoO_{3-x} phototransistors is experimentally supported by the temperature-dependent transport measurements as well as the controlled photocurrent measurements on devices with different electrodes. As a result of this novel mechanism, a record-high detectivity of 9.8×10^{16} Jones with a reasonably ultrafast photoresponse in 2D phototransistors has been achieved. The devices were systematically characterized at different gate voltages, light wavelengths, and incident power, and the statistical analysis further confirms the stable nature of device performance. As a complement, I would like to raise some minor points needed to be further addressed, and thus recommend the publication of this work in high-impact journals such as Nature Communications.

Response:

We thank the reviewer for his/her positive and insightful comments, which are essentially important to further improve our manuscript. We carefully revised the manuscript and addressed all the concerns of the reviewer in the revised manuscript.

Details are shown as follows.

1. The qualitative description of PIBL process in Page 11 (line 9-15) is still unclear. The authors should explain why the separation of photo-generated charge carriers at source heterojunction can lead to the increase of source Fermi-level (actually the increase of entire band), is it due to the accumulation of photo-generated holes at source interface? In addition, the decrease of energy barrier at source interface is not explicitly demonstrated in Fig. 4e and i, where the barrier height seems to remain or even increase due to the larger band bending.

Response:

Thank you very much for the valuable comments. We summarized the details to clearly clarify the PIBL process:

Firstly, we would like to point out light illumination does not directly change the bulk Fermi level (entire channel). As the reviewer stated, the accumulated photo-carriers combined with drift motion results in the increasing Fermi level. Our device is generally operated at OFF state of the photo-transistor, where MoS₂ is almost fully depleted. At this region, MoS₂ present very small density of states (DOS). By light illumination, electron-hole pairs are dynamically generated at the reversely biased source junction and subsequently separated by the built-in field. Or in other words, there is an equivalent charge transfer between α -MoO_{3-x} and MoS₂, which rapidly modifies the Fermi level at the source and leverages the carrier density. Please note that the source-drain bias drives the electrons injected into MoS₂ to move across the whole channel. This drift transport process increases the channel carrier density and changes the bulk Fermi level due to the small DOS.

Secondly, we thank the reviewer for correcting the figure flaws. We thus revised the figure to highlight the band lowering as well as the narrowing of space charge region by which the photo-responses are significantly enhanced. The revised Fig. 4e and i in the main text, are also shown in **Fig. R1**.

Figure R1 Double-heterojunction photo-induced barrier lowering mechanism. **a** Energy band diagram at the light-on state to illustrate the photo-induced barrier lowering mechanism. The blue symbol (1) represents the photo-generation of electron-hole pairs, the red symbol (2) the holes and electrons separating process to reduce barrier and increase electrons injection, and the green symbol (3) the injection electrons reducing the barrier at drain side. **b** Energy band diagram of device T3. When there is incident light, a large photo gain was achieved because Source barrier lowering enhanced electron injection. E_C is the conduction band minimum, E_T is the defect band and E_V is the valence band maximum.

Corresponding modification has been made in Page 11 (Line 22), Page 12 (Lines 1-2, 11-15) in blue color in the revised manuscript.

2. In addition to the band structures of α -MoO_{3-x} and MoS₂, the n-type doping of MoS₂ channel by HfO₂ capping layer is also important to narrow the interfacial energy barrier for electron injection (Supplementary Fig. 7), which should be highlighted in

the main text, e. g. Page 10, paragraph 2.

Response:

We fully agree with the reviewer that we should highlight the importance of HfO₂ capping layer for narrowing the interfacial energy barrier. we added the sentence “which narrows the interfacial energy barrier between α -MoO_{3-x} and MoS₂” in the main text.

Corresponding modification has been made in Page 4 (Line 21) in blue color in the revised manuscript.

3. The extraction of noise power density S in the dark around the off state ($V_G \sim$ from -30 to -40 V) is not clearly demonstrated in the main text Page 5 and SI Supplementary Fig. 9. In general cases, S should be experimentally determined by the noise measurements in Supplementary Fig. 8 g. Does the authors mean that the real S of the device around off state is covered by the background noise, which can be further extracted by the proportional relation of S to I_2 for flick noise? The authors should carefully depict how to extract the S from Supplementary Fig. 8 g and h to obtain Fig. 2a, since S is the key parameter for determining the detectivity. In addition, the Fig. 8g, h and Fig. 9 in SI seems to present the same data for two different devices. It is recommended to remove Fig. 9 and add the corresponding discussion for Fig. 8g and h.

Response:

Thank you very much for the valuable comments. As the reviewer pointed out, S

is the key parameter for determining the detectivity. It is necessary to explain how to extract the S in the dark around the OFF state ($V_{GS} \sim$ from -30 to -40 V) clearly. **Figure R1a** shows the relationship between current noise and frequency at various back gate voltages. All these low-noise spectra exhibit a typical $1/f$ power density. Here, we must notice that the S around the OFF state ($V_{GS} \sim$ from -30 to -40 V, in blue and red) drowns with background noise (in black). It is well-known that the $1/f$ (flicker) noise is mainly dominated by fluctuations of carrier density or mobility. The current I_{Dark} was extracted from the $I_{DS}-V_{GS}$ characteristics in the dark (**Fig. R1b**), and **Fig. R1c** shows that the noise power spectral density $S(f)$ is proportional to I_{Dark}^2 , indicating that the photocurrent does not determined by external fluctuations such as interfacial traps (Rev. Mod. Phys. **53**, 497–516, 1981). At $f=1$ Hz, the mean value of S/I_{Dark}^2 is about 4×10^{-7} Hz⁻¹ (**Fig. R1c**), so that we can obtain the real S of the device around the OFF state by $S=4 \times 10^{-7} \times I_{Dark}^2$ (**Fig. R1d**). When the device is in the OFF state ($V_{GS} = -35.2$ V), the S of the device can be as low as 9.7×10^{-32} A²/Hz.

In order to let readers clearly understand the extraction of noise around the off state of the device, we move the Fig. 9 in SI to the main text and add the corresponding discussion.

Figure R2 Characterization of noise power density (S) of the phototransistor. **a** S as a function of frequency at $V_{DS} = 1$ V with different V_{GS} of 0 V, -15 V, -20 V, -25 V, -30 V, -35 V as well as the background noise from top to down. S decreases rapidly with a decrease of V_{GS} , and drowns with background noise when the V_{GS} decreases to be -30 V. **b** Transfer curves of this device at $V_{DS} = 1$ V in dark. **c** Normalized noise power density (S/I_{Dark}^2) as a function of frequency at $V_{DS} = 1$ V with different V_{GS} of 0 V, -15 V, -20 V, -25 V. **d** S as a function of V_{GS} at $V_{DS} = 1$ V.

Corresponding modification has been made in Page 5 (Lines 14-16) and Page 6 (Lines 1-5) in blue color in the revised manuscript and Page 10 (Lines 12-25) and Page 11 (Line 1) in blue color in the revised supplementary materials.

4. The language of this manuscript is required to be polished to improve the readability for a broad audience. Several expressions are not sufficiently accurate and concise for

NPG publications, and a lot of spelling mistakes and grammar errors have been found throughout the manuscript. For example:

Page 2 (line 8): “Here, we report an (ultrasensitive) molybdenum-based”;

Page 3 (line 3): “pure atomically thick 2D layer materials” should be revised to “atomically-thin 2D layered materials”;

line 8 and 11: “(and) however”;

Page 5 (line 14): “form”, Page 6 (line 1): “deceases”;

Page 8 (line 13 and 14): “increased” and “increases” (tense should be consistent);

Page 10 (line 2): “experimental data such as absorbance spectrum” should be revised to “experimental characterizations such as absorption spectroscopy”;

Page 11 (line 5): “which blocks the traps whose energy is below the barrier” should be revised to “which blocks the traps with the energy below the barrier”;

Fig. 4b: “3.05 (eV)”.

SI Page 11 (line 14): “indicating that the photocurrent does not determined by external” should be revised to “indicating that the photocurrent is not determined by external”.

Response:

We are grateful for the valuable comments. We have polished our language and corrected the spelling and grammar mistake of our manuscript.

Corresponding modification has been made in Page 2 (Line 8), Page 3 (Lines 3, 8 and 11), Page 6 (Lines 4 and 8), Page 9 (Lines 14-16 and 18), Page 11 (Line 2) Page 12 (Lines 8-9) in blue color and add “eV” in Fig. 4b in the revised manuscript and Page 10 (Line 17-18) in the revised supplementary materials.

5. Several relevant references for photogating mechanism should be added into the introduction:

1) *Adv. Opt. Mater.* **8**, 1901971 (2020)

2) *Adv. Sci.* **7**, 2002393 (2020)

Response:

We have added these two important references to Ref 27 (*Adv. Sci.* **7**, 2002393 (2020)) and Ref 40 (*Adv. Opt. Mater.* **8**, 1901971 (2020)) in the revised manuscript.

Reviewer #2 (Remarks to the Author):

In the manuscript “An ultrasensitive molybdenum-based double-heterojunction phototransistor”, the authors present a high-performance photodetector based on MoS₂ flakes contacted through two MoO₃ electrodes, and account for the high performances on the basis of a novel mechanism involving the lowering of Schottky barriers under light irradiation. The proposed mechanism is in agreement with all the presented data, so it appears reasonable. In the present version, I particularly appreciate the Supplementary figures summarizing the performances of the different devices (Supp Fig. 12 and Supplementary Table1).

I find a major issue in the manuscript, in the extraction of the phototransistor performance. The authors do measure the spectral noise at the different back gate

voltages, but they do not use the experimentally measured data to evaluate the detectivity. By contrast, they try to estimate the “intrinsic” noise level and use it for the calculation of the detectivity. The details of how the intrinsic value is extracted are not completely clear to this reviewer, as the authors only mention “The noise density spectral (S) as a function of gate voltage (VGS) at $f = 1$ Hz is shown in Fig. 2a, which is inferred from the normalized noise power density (S/I_{dark}^2) as a function of frequency” (pag 5, lines 13,14).

In my understanding, the authors notice that S is proportional to I^2 (in dark) at high gate voltages VG, and make the assumption that the same S vs I^2 holds for low VG, where S is below the instrumental sensitivity. Therefore, they “infer” the intrinsic S at low VG by multiplying one of the (S/I^2) traces measured at high VG by the dark current IDS measured at low VG. By doing this, they obtain the graph shown in the response to the referee (Fig. R4d, and not shown in the last version of manuscript or SI), where S decreases from 10^{-30} A²/Hz at 1 Hz to 10^{-35} at high frequency.

The underlying assumption that the spectral shape of S does not depend on VG is incorrect. Indeed, the graph shown in R4d does not make sense, since at high frequency the noise level is below the intrinsic shot noise, which I estimate to be approx 10^{-32} or 10^{-33} A²/Hz (frequency independent).

(Question: have I understood correctly how the "intrinsic" noise was estimated?)

The authors should describe this procedure with more details, and show the figure R4d in SI, with the caveat that the Fig. R4d would only be valid at low frequencies.)

The authors should calculate the detectivity using the measured S value,

highlighting how the (measured) detectivity underestimates the real device detectivity. The inferred detectivity value (as calculated now) could be provided as an estimation of the intrinsic value, but the measured value should be also given in abstract, introduction, Fig 2a and Fig. 2e.

In any case, even considering the measured detectivity, the performances compare positively with previous works (their star would remain in the top left corner of Fig. 2g, although closer to the cloud of points). As a notice, in Fig. 2g several works estimate S as the shot noise; so the overestimation is even worse than in the present case. Perhaps the authors could plot in different ways the works in which the noise is measured or estimated, and perhaps they could add for their case both the measured and the estimated value – so the comparison could be “fair”.

And as another minor point, the authors now show the temperature dependence of the IV traces, from which they extract the Schottky barrier height. It is not clear whether this dataset was measured for a MoO/MoS₂/metal or MoO/MoS₂/MoO device; but they treat the data as if there was only one Schottky barrier, which goes against the mechanism put forward to account for the measured data. I suppose that this could still be a good approximation, considering that most of the effect comes from one of the two interfaces; the authors might want to comment on this.

Overall, in my opinion, the data presented in this work could be suitable for Nature Communications, but the performance extraction should be carefully clarified before I can recommend the acceptance of this work.

Response:

We thank the reviewer for his/her positive and insightful comments, which are essentially important to further improve our manuscript. We carefully revised the manuscript and addressed all the concerns of the reviewer in the revised manuscript. Details are shown as follows.

As both reviewers pointed out that the details of how the intrinsic value was extracted are not completely clear, so that it is necessary for us to further explain it. **Figure R3a** shows the relationship between current noise and frequency at various back gate voltages. All these low-noise spectra exhibit a typical $1/f$ power density. Here, we must notice that the S around the OFF state ($V_{GS} \sim$ from -30 to -40 V, in blue and red) drowns with background noise (in black). It is well-known that the $1/f$ (flicker) noise is mainly dominated by fluctuations of carrier density or mobility. The current I_{Dark} was extracted from the $I_{DS}-V_{GS}$ characteristics in the dark (**Fig. R3b**), and **Fig. R3c** shows that the noise power spectral density $S(f)$ is proportional to I_{Dark}^2 , indicating that the photocurrent does not determined by external fluctuations such as interfacial traps (Rev. Mod. Phys. **53**, 497–516, 1981). At $f = 1$ Hz, the mean value of S/I_{Dark}^2 is about 4×10^{-7} Hz^{-1} (**Fig. R3c**), so that we can obtain the intrinsic S of the device around the OFF state by $S = 4 \times 10^{-7} \times I_{Dark}^2$ (**Fig. R3d**). When the device is in the OFF state ($V_{GS} = -35.2$ V), the S of the device can be as low as 9.7×10^{-32} A^2/Hz .

In addition, according to the reviewer's suggestions, we have calculated the detectivity using the measured S value. The value of measured S at $V_{GS} = -30$ V is about 6×10^{-26} A^2/Hz ($f = 1$ Hz), the maximum R at $V_{GS} = -30$ V is about 2.6×10^5 A/W ($P_{in} = 0.46$ mW/cm²), and the area of this device is about ~ 2.5 μm^2 . Using the equation

$D^* = (AB)^{1/2} R / S^{1/2}$, we can calculate the detectivity of this device about 1.7×10^{14} cm $\text{Hz}^2 \text{W}^{-1}$. However, the measured S at $V_{\text{GS}} = -30$ V is drowned with background noise (Fig. R3a), as a result, the measured detectivity underestimates the real device detectivity. The inferred detectivity value and the measured value are both given in abstract, introduction, Fig 2a and Fig. 2e. And we also provide the Fig. R4d in previous response letter in the revised supplementary materials.

Finally, we need to clarify that we measured the temperature dependence of the $I_{\text{DS}}-V_{\text{GS}}$ traces in a $\alpha\text{-MoO}_{3-x}/\text{MoS}_2/\alpha\text{-MoO}_{3-x}$ device. In fact, there are two junctions in this structure. We fully agree with the reviewer that this could still be a good approximation, considering that most of the effect comes from one of the two interfaces.

Figure R3 Characterization of noise power density (S) of the phototransistor. **a** S as a function of frequency at $V_{\text{DS}} = 1$ V with different V_{GS} of 0 V, -15 V, -20 V, -25 V, -30

V, -35 V as well as the background noise from top to down. S decreases rapidly with a decrease of V_{GS} , and drowns with background noise when the V_{GS} decreases to be -30 V. **b** Transfer curves of this device at $V_{DS} = 1$ V in dark. **c** Normalized noise power density (S/I_{Dark}^2) as a function of frequency at $V_{DS} = 1$ V with different V_{GS} of 0 V, -15 V, -20 V, -25 V. **d** S as a function of V_{GS} at $V_{DS} = 1$ V.

Corresponding modification has been made in Page 2 (Lines 11-15), Page 3 (Line 22), Page 4 (Lines 1-4), Page 5 (Lines 14-16) and Page 6 (Lines 1-5 and 17-18), Page 9 (Lines 2-4) in blue color and Figs 2a, 2e and 2g in the revised manuscript and Page 10 (Lines 9 and 12-25), Page 11 (Line 1) Page 24 (Lines 8-11) in blue color and Fig. S8i in the revised supplementary materials.

REVIEWERS' COMMENTS

Reviewer #1 (Remarks to the Author):

In this revised manuscript, the authors have addressed the previously-raised minor points. Three more suggestions: (1) The term of "real noise" throughout the manuscript is recommended to be replaced by "intrinsic noise", and on line 13 Page 2, the word "true" should be removed. (2) In Fig. 2a, the "black" background noise is suggested to be moved to the top layer to make it more obvious (similar to Fig. S8g). Moreover, the voltage increasing step and the description of Fig. 2e inset should be added into the figure caption, and both symbols of hollow and solid polygon should be displayed at the right side of Fig. 2g. (3) The expression how to extract the intrinsic current noise (Page 6, line 3) is still not very clear for readers in the main text. It is recommended to move some description in SI (e. g. Fig. S8) here to further improve the readability. Thus, I would recommend the publication of this work in Nature Communications.

Reviewer #2 (Remarks to the Author):

The authors have addressed my comments in a satisfactory way. They now provide the measured and estimated noise level, and they compare their performance to previous works in a more transparent way. In my opinion, the quality of the work has improved significantly during the reviewing process, and I am glad to recommend the publication of this work in Nature Communications.

Responses to the comments of reviewers

Reviewer #1 (Remarks to the Author):

In this revised manuscript, the authors have addressed the previously-raised minor points. Three more suggestions: (1) The term of “real noise” throughout the manuscript is recommended to be replaced by “intrinsic noise”, and on line 13 Page 2, the word “true” should be removed. (2) In Fig. 2a, the “black” background noise is suggested to be moved to the top layer to make it more obvious (similar to Fig. S8g). Moreover, the voltage increasing step and the description of Fig. 2e inset should be added into the figure caption, and both symbols of hollow and solid polygon should be displayed at the right side of Fig. 2g. (3) The expression how to extract the intrinsic current noise (Page 6, line 3) is still not very clear for readers in the main text. It is recommended to move some description in SI (e. g. Fig. S8) here to further improve the readability. Thus, I would recommend the publication of this work in Nature Communications.

Response:

We thank the reviewer for his/her insightful suggestions, which are essentially important to further improve our manuscript. We carefully revised the manuscript and addressed all the concerns of the reviewer in the revised manuscript. Details are shown as follows.

First, we used the term of “intrinsic noise” to replace the term of “real noise” throughout the manuscript and removed the word “true”. Second, we moved the “black” background noise to the top layer to make it more obvious in Fig. 2a and add the voltage increasing step into the figure caption. We also added the description of Fig. 2e inset into the figure caption and displayed solid polygon at the right side of Fig. 2g. Finally, we moved some description about how to extract the intrinsic current noise in SI to the main text.

Corresponding modification has been made in Page 4 (Lines 1 and 3), Page 6 (Lines 5, 7-9), Page 8 (Lines 2-4, 8), Page 9 (Line 1-2) in blue color and Fig. 2g in the revised manuscript.

Reviewer #2 (Remarks to the Author):

The authors have addressed my comments in a satisfactory way. They now provide the measured and estimated noise level, and they compare their performance to previous works in a more transparent way. In my opinion, the quality of the work has improved significantly during the reviewing process, and I am glad to recommend the publication of this work in Nature Communications.

Response:

Thanks for your important comment.